



# Seasonal study of the Small-Scale Variability of Dissolved Methane in the western Kiel Bight (Baltic Sea) during the European Heat Wave in 2018

Sonja Gindorf[1,2,*], Hermann W. Bange[1], Dennis Booge[1], and Annette Kock[1,3]

[1] Marine Biogeochemistry, GEOMAR Helmholtz Centre for Ocean Research Kiel, Kiel, Germany

[2] now at: Department of Environmental Science, University of Stockholm, Stockholm, Sweden

[3] now at: Landesamt für Landwirtschaft, Umwelt und ländliche Räume, Flintbek, Germany

* corresponding author: sonja.gindorf@aces.su.se

## Abstract

Methane ($CH_4$) is a climate-relevant atmospheric trace gas which is emitted to the atmosphere from coastal areas such as the Baltic Sea. The oceanic $CH_4$ emission estimates are still associated with a high degree of uncertainty partly because the temporal and spatial variability of the $CH_4$ distribution in the ocean surface layer is usually not known. In order to determine the small-scale variability of dissolved $CH_4$ we set up a purge-and-trap system

with a significantly improved precision for the $CH_4$ concentration measurements. We measured the distribution of dissolved $CH_4$ in the water column of the western Kiel Bight and Eckernförde Bay in June and September 2018. The top 1 m was sampled in high-resolution to determine potential small-scale $CH_4$ concentration gradients within the mixed layer. $CH_4$ concentrations throughout the water column of the western Kiel Bight and Eckernförde Bay

were generally higher in September than in June. The increase of the $CH_4$ concentrations in the bottom water was accompanied by a strong decrease in $O_2$ concentrations which led to anoxic conditions favorable for microbial $CH_4$ production in September. In summer 2018, northwestern Europe experienced a pronounced heatwave. However, we found no relationship between the anomalies of water temperature and excess $CH_4$ in both the surface

and the bottom layer at the site of the Boknis Eck Time-Series Station (Eckernförde Bay). Therefore, the 2018 European heatwave most likely did not affect the observed increase of the $CH_4$ concentrations in the western Kiel Bight from June to September 2018. The high-resolution measurements of the $CH_4$ concentrations in the upper 1 m of the water column



were highly variable and showed no uniform decreasing or increasing gradients with water

depth. Overall, our results show that the CH$_4$ distribution in the water column of the western Kiel Bight and Eckernförde Bay is strongly affected by both large-scale temporal (i.e. seasonal) and small-scale spatial variabilities which need to be considered when quantifying the exchange of CH$_4$ across the ocean/atmosphere interface.

# 1. Introduction

Methane (CH$_4$) is an important atmospheric greenhouse gas that is produced in open ocean and coastal environments (see e.g. Reeburgh, 2007; Wilson et al., 2020). Oceanic CH$_4$ emissions depend on the interplay of various biogeochemical, oceanographic and biological factors that drive production, consumption and transport processes of CH$_4$ (see e.g. Bakker et al., 2014). Oceanic CH$_4$ can be either of geologic or biological origin (see e.g. Bakker et

al., 2014; Reeburgh, 2007; Wilson et al., 2020). Generally, open ocean surface waters are at atmospheric equilibrium or slightly oversaturated (Weber et al., 2019). Oceanic emissions (incl. open and coastal areas) contribute only ~1-3% to the global CH$_4$ budget (Saunois et al., 2020). Coastal areas including shelves and estuaries account for up to 75% of total CH$_4$ oceanic emissions to the atmosphere (Weber et al., 2019). However, large uncertainties

remain regarding the temporal and spatial variability of CH$_4$ concentrations and the CH$_4$ emissions to the atmosphere (see e.g. Weber et al. 2019). Moreover, temporary extreme events such as warming of the upper ocean due to heatwaves can affect the dissolved CH$_4$ concentrations and its emissions (Borges et al., 2019; Humborg et al., 2019). To this end, we present here a seasonal study of dissolved CH$_4$ gradients the western Kiel Bight (incl.

Eckernförde Bay) during the European heatwave in 2018. The overarching objectives of this study were to (i) to set up a CH$_4$ measurement system with a precision that allows detection of small-scale variability of CH$_4$ concentrations, (ii) decipher the small-scale variability of dissolved CH$_4$ in the upper water column on a seasonal basis, (iii) assess how extreme events such as the European heatwave in 2018 might affect the CH$_4$ concentrations and (iv)

determine the consequences of CH$_4$ concentration gradients for the CH$_4$ emissions to the atmosphere.

## 1.1 Study Site

The western Kiel Bight and the Eckernförde Bay are affected by the inflow of water along the bottom, from the North Sea through the Kattegat and the Great Belt, and also the surface

outflow of brackish water (Bange et al., 2010; Bange et al., 2011; Lennartz et al., 2014; Steinle et al., 2017); this results in strong fluctuations in bottom water salinity, between 17 and 24 (Lennartz et al., 2014). Complete vertical mixing of the water column is prevented



from March to September, as a strong pycnocline develops due to the surface warming and the distinct salinity gradient between inflowing and outflowing water masses. During the

winter months, the whole water column is mixed as a consequence of storms and surface-water cooling (Bange et al., 2010). Strong phytoplankton blooms in early spring (February–March) and autumn (September–November) are followed by high rates of organic matter sedimentation and microbial respiration, and thus also the consumption of $O_2$ (Bange et al., 2010; Dale et al., 2011). Consequently, pronounced hypoxia and sporadic anoxia occur in

the bottom waters during late summer (Bange et al., 2010; Lennartz et al., 2014; Steinle et al., 2017). The occurrence of anoxic events has been continuously increasing in frequency since the 1970s (Lennartz et al., 2014). The high sedimentation rates of organic matter favour methanogenesis in the muddy sediment, and $CH_4$ release to the water column, resulting in high $CH_4$ concentrations in the overlying water (Bange et al., 2010; Ma et al.,

80    2019).

## 2. Methods

Here, we present $CH_4$ measurements from two research cruises with the R/V Alkor as part of the Baltic GasEx experiment in 2018. The cruises AL510 (Booge, 2018) and AL516 (Booge, 2019) took place in June 2018 and September 2018, respectively. $CH_4$ samples were taken

from 9 (AL510) and 10 (AL516) CTD rosette casts (Figure 1). The Baltic GasEx experiment was as a dual-tracer experiment to investigate the air-sea gas exchange in the western Kiel Bight (Ho et al., 2019). To this end, the cruise tracks emerged from following the patch of a surface water mass marked with a pair of tracers ($^3$He/$SF_6$) which were released at the start of each campaign and led to a high spatial resolution coverage of the western Kiel Bight. To

study the vertical $CH_4$ distribution within the water column, samples were taken from the mixed layer, from within the pycnocline, and from the water below the pycnocline (Booge, 2018, 2019). The CTD was mounted to the rosette water sampler with twelve 10 L Niskin bottles that were closed during the upcast at the requested sampling depths.

To examine potential $CH_4$ concentration gradients in the very near-surface waters in higher

resolution, additional samples from a zodiac were taken at selected stations. To avoid turbulence distributions caused by the ship, the sea water samples were taken at some distance from the ship. Using a self-built sampling device that consisted of an aquarium pump attached to a floating board, water from 0.1, 0.5 and 1 m depth below the surface was pumped on board the zodiac as described in detail by Fischer et al. (2019). In addition,

discrete underway (UW) samples were taken from the ship's continuous seawater supply system with a water inlet at ~2 m depth.



Triplicate water samples were taken through a silicon hose connected to the Niskin bottle (CTD samples), aquarium pump (zodiac samples), or the ship's underway system (UW samples). At a low flow-rate, 20 mL amber glass vials were filled bubble-free by overflowing

the approximate threefold volume of seawater. The vials were closed with butyl rubber stoppers and crimp-sealed with aluminium caps. To inhibit microbial activity, 50 µL of saturated mercury chloride solution ($HgCl_2$ (aq)) were added to each sample. To compensate for the added volume of $HgCl_2$ solution, a needle with a 3 mL syringe body was inserted into each sample before $HgCl_2$ injection. The samples were stored at room temperature in the

dark until the measurements were carried out.

## 2.1 Purge and Trap System

We used a self-built purge and trap (PT) system coupled to a gas chromatograph equipped with a flame ionization detector (GC-FID) to measure $CH_4$ in the surface water and the water column. The set up (Figure 2) of the PT measurement system (Figure 2A) can be divided into

three sections that describe the purge unit (Figure 2B), the trapping unit (Figure 2C), and the GC-FID system (Figure 2D). The materials that were used are listed in the supplement. Helium (He) is used as the purge gas and as the carrier gas for the (GC). The gas stream is split and directed through the purge unit while a continuous gas stream through the GC is maintained. A digital thermometer is installed next to the system to monitor the temperature.

The purge unit contains a 2-position 4-way-valve that can be switched to enable the purging of the sample ('purge' mode) or the emptying of the purge chamber ('waste' mode). In the purge mode, the He gas stream is directed through the sample vial and consecutively through the purge chamber. The sample water is pushed into the purge chamber when the He gas stream is turned on. A long needle that reaches the bottom of the sample vial is used

to completely empty the vial. Backflushing of the water into the sample vial is restricted by two check valves at the hose between the purge chamber and the sample. The purge chamber consists of a 50 mL sample vial that is placed upside down to minimize leakage through the stopper. The purge flow is directed into the purge chamber through a short needle with the needle tip placed close to the stopper to ensure the purging of the entire

water sample. When the He gas is bubbling through the sample, the dissolved gases are stripped from the water phase. Due to its low solubility in seawater at room temperature and normal pressure (Duan et al., 1992), $CH_4$ is stripped out within 4 minutes using a purge flow of approximately 0.03 L min$^{-1}$. The gas is extracted from the purge chamber via a long needle with the needle tip placed close to the bottom of the upside-down vial. The gas is then

directed through a filter and dried over a Nafion™ dryer with a counterflow of dry compressed air at a flow rate of approximately 200 mL min$^{-1}$. Additionally, two glass tubes



filled with phosphorus pentoxide (Sicapent®; $P_2O_5$) are used to further dry the gas stream. To ensure a continuous and uniform flow rate during the measurements, a flowmeter is installed before the gas is transferred to the trapping unit.

The He gas stream from the purge unit as well as the carrier gas stream of the GC are connected to a two-position-six-port valve that enables the switching of the gas stream through the $CH_4$ trap. In the 'trap' position, the purge gas is conducted through the trap, which consists of a 20 cm x 1/8" stainless steel column filled with Spherocarb (100-200 mesh). The trap is put into liquid nitrogen during the trapping procedure. In the 'desorb'

position, the GC carrier gas flow is conducted through the trap, which is subsequently removed from the liquid nitrogen and put into a water bath of ~90°C to desorb the trapped gases from the column. The gas flow is then directed through the GC-FID system (D).

(D) The GC-FID used in this setup is an HP 5890 II GC equipped with an FID detector. The gases are separated over a 1/8" 6' stainless steel column, with a carrier gas flow of ~30

mL min$^{-1}$. A similar flow rate of the carrier gas streams during the trapping and the desorption steps was chosen to avoid baseline shifts when switching the Valco valve.

## 2.2 Calibration

On each measurement day, a set of standard gas mixtures and blank measurements with and without the injection of 20 mL He was measured prior to sample measurements. The gas

standards have been calibrated against two National Oceanic and Atmospheric Administration (NOAA) primary standards that were provided by the SCOR Working Group 143 for the intercomparison of oceanic $CH_4$ and $N_2O$ measurements (Wilson et al., 2018). Prior to standard measurements, one sea water sample was purged on every measurement day. This water was left inside the purge chamber and used for the blank and standard

measurements. For the standard and He blank injection, 20 mL plastic syringes were used. A Safeflow® infusion valve was attached to the check valve at the standard injection port to reduce the dead volume of the injection port. After each standard injection, 3 mL of He was injected with a 3 mL plastic syringe through the port to ensure that all injected volume of the standard is injected into the purge chamber. The He flow through the purge unit is switched

off during the standard injection. Different volumes of the standard were injected into the purge system to create a calibration curve that covered the full concentration range of the samples. The amount of the injected standards was calculated from the respective mole fraction and the injected volume using the ideal gas law. The chromatography software Chromstar 6.3 (SCPA GmbH, Weyhe-Leeste, Germany) was used for data acquisition and

manual integration of the $CH_4$ peaks.





### 2.3 Comparison of Static Headspace Equilibrium and Purge and Trap

To ensure that the PT measurements are comparable with measurements with the previously used static headspace equilibration (HS) method (see e.g. Ma et al., 2020), triplicates of seawater samples from the six standard depths were taken for PT and HS during a Boknis Eck cruise. The $CH_4$ concentrations ranged from 5 to 222 nmol $L^{-1}$ allowing a comparison over a broad concentration range (Figure 3). Over all depths, the PT measured concentrations were slightly lower and showed significantly less variation among the triplicates, thereby reflecting a better precision of the PT measurements over the HS method (Table 1). The direct comparison of both techniques shows that the measurements agree well with the HS measurements (Figure 4). Other studies have proven higher precision and sensitivity as well as handling benefits of PT over HS (e.g. Capelle et al., 2015).

### 2.4 Calculation of Dissolved $CH_4$ Concentrations

The $CH_4$ concentration in the sample was calculated from the linear calibration fit using equation (1). The mean area of the blank measurements was subtracted from the sample peak area to account for the background contamination of the system.

$$n = \frac{PA_{sample} - PA_{blank}}{\delta} \ (1),$$

where       $n$ is the amount of $CH_4$ [nmol] in the sample,

         $PA_{Sample}$ is the peak area of the measured sample,

         $PA_{Blank}$ is the mean peak area of the measured blanks,

         $\delta$ is the slope of the calibration curve [$nmol^{-1}$].

The $CH_4$ concentration c [nmol $L^{-1}$] was calculated as the ratio between n and the sample volume V [L]. V was determined experimentally to be 0.0203 ± .0002 L.

We estimated the standard deviation for triplicates or duplicates according to the statistical analysis of David (1951). The mean analytical error of the $CH_4$ concentration was +/- 5.7 and 3.1 % during AL 510 and AL 516, respectively.

### 2.5 ΔCH₄ and CH₄ Saturations

Temperature and salinity data from the ship's thermosalinograph were used to compute the excess of $CH_4$ ($\Delta CH_4$) as the difference between the measured $CH_4$ concentration (c, see above) and the $CH_4$ equilibrium concentration ($c_{eq}$). The solubility equation for $CH_4$ in seawater (Wiesenburg and Guinasso, 1979) was applied to calculate $c_{eq}$ (in nmol $L^{-1}$). The atmospheric dry mole fraction of $CH_4$ was taken from records of the Mace Head observatory in Ireland, which is part of the Advanced Global Atmospheric Gases Experiment (AGAGE,



https://agage.mit.edu/). Monthly means of 1918.24 ppb ± 0.2% for June and 1925.83 ppb ±
0.2% for September 2018 (Dlugokencky, 2020) were used for the calculations.

$CH_4$ saturations ($CH_4$sat in %) were computed as

$$CH_4\text{sat} = 100 * c/c_{eq} \text{ (2).}$$

## 2.6 Oxygen Measurements

During both cruises, a CTD-mounted altimeter (sn#453) oxygen sensor was used to obtain
CTD-$O_2$ profiles from the surface to 1 m above the bottom. Additionally, 112 and 105
discrete oxygen samples were Winkler titrated during AL 510 and AL 516, respectively, and
were used to calibrate the sensor data.

## 2.7 Temperature and $CH_4$ Anomalies

The measurements at the Boknis Eck Time-Series Site performed during this study allows us
determine the effect of the European heatwave in 2018 in the context of the monthly time-
series measurements of water temperature and dissolved $CH_4$ at Boknis Eck (Lennartz et al.,
2014; Ma et al. 2020). To this end, we computed the anomalies of water temperature and
$\Delta CH_4$ in 1 m and 25m water depth for the period of January 2006 to December 2018. The
anomalies were defined as

$$\Delta T = T - T_{i, \, avg} \text{ (3) and}$$

220                   $$\Delta \, (\Delta CH_4) = (\Delta CH_4) - (\Delta CH_4)_{i, \, avg} \text{ (4),}$$

where T is the measured monthly water temperature (T) in the period January 2006 to
December 2018 in 1 (25) m depth at Boknis Eck (Lennartz et al., 2014). $T_{i, \, avg}$ is the mean
water temperature in 1 (25) m depth of the respective month i over this period at Boknis Eck .
The resulting $\Delta T$ is the anomaly of the water temperature which is cleaned from seasonal
differences throughout each year. $\Delta \, (\Delta CH_4)$ is calculated similarly to $\Delta T$ in the same time
period using $\Delta CH_4$ which is the monthly excess $CH_4$ ($\Delta CH_4$, see above) in 1 (25) m depth.
$\Delta CH_4$ was computed as the difference of the monthly measurements of dissolved $CH_4$ in 1
(25) m water depth (Ma et al., 2020) and the monthly $C_{eq}$ (see above) in 1 (25) m water
depth. $C_{eq}$ was calculated according to the solubility equation of Wiesenburg and Guinasso
(1979) with the monthly water temperature and salinity at Boknis Eck in 1 (25) m depth
(Lennartz et al., 2014) and the monthly atmospheric dry mole fractions of $CH_4$ measured at
Mace Head AGAGE observatory (https://agage.mit.edu/) from January 2006 to December
2018. Extremely high $CH_4$ surface concentrations with concentrations in the range from 87 to
689 nmol $L^{-1}$ have been measured in four months (November 2013, February/March 2014



and December 2014) and were, thus, omitted from the data set to avoid a statistical bias of
the data set.

## 3. Results and Discussion

### 3.1 June 2018

The hydrographic conditions during AL 510 in June 2018 revealed a strong stratification of
the water column: while the upper ~10 m were comparably uniform, a strong gradient in
temperature prevailed between 10 and 15 m, and the lower water column (15-20 m) showed
a strong salinity gradient (Figure 4). This is in line with former studies in the respective area
(e.g. Bange et al., 2010; Dale et al., 2011; Lennartz et al., 2014; Ma et al., 2019, 2020).

Between June 9 and 13, surface water temperatures exceeded 20 °C, which is higher than
the average surface temperature measured at the Boknis Eck station in June (Lennartz et al.,
2014).

The highest $O_2$ concentrations were measured between approximately 7 m and 18 m depth
and between the 1009 and the 1012 kg m$^{-3}$ isopycnals. At the beginning of the
measurements, the $O_2$ maximum had a vertical extension of more than 10 m, which
decreased continuously in its thickness and intensity until it vanished after the 15[th] of June.
The whole water column was oxygenated throughout the cruise, with oxygen concentrations
decreasing to ~120 µmol L$^{-1}$ in the bottom waters. At this time of the year, the oxygen
depletion in the deep water starts to evolve (Lennartz et al., 2014). The water column $CH_4$
concentrations ranged from 2.8 nmol L$^{-1}$ (99 % saturation) in the surface waters to
28.3 nmol L$^{-1}$ (750 %) in the bottom waters.

### 3.2 September 2018

During the AL 516 campaign in September 2018, the water column also showed a strong
stratification below 10 m with pronounced gradients in temperature, salinity and $O_2$
concentrations (Figure 5). In contrast to the AL 510 cruise in June 2018, the stratification
below 10 m seemed to be primarily driven by the salinity gradient. The surface water showed
a comparably homogeneous distribution of about 17 °C down to the 1014 kg m$^{-3}$ isopycnal at
~15 m. A strong temperature decrease was observed below 20 m, with temperatures ~12°C
in the bottom waters. An increased surface density at the beginning and the end of the cruise
(before Sept. 14, after Sept. 22) indicated the upwelling of waters from 5-10 m to the surface.
A strong storm event on the 21[st] could have induced this upwelling (mean wind speed on 21[st]
~12 m s$^{-1}$). The bottom water salinity was lower in the beginning than in the end of the cruise





and the higher salinities were shifted upward in the water column in agreement with the upshift of the 1016 kg m$^{-3}$ isopycnal.

During the whole cruise, the highest $O_2$ concentrations were measured in the surface waters
above the 1013 kg m$^{-3}$ isopycnal. Between the 1014 and 1015 kg m$^{-3}$ isopycnals a strong oxycline could be observed as $O_2$ decreased by approximately 100 µmol L$^{-1}$ within a few meters. With increasing depth, the $O_2$ concentration decreased to suboxic ($O_2 < 5$ µmol L$^{-1}$) and almost anoxic ($O_2 \sim 0$ µmol L$^{-1}$) conditions in the bottom water. However, during AL 516 no indication of sulfidic conditions (e.g. the smell of hydrogen sulphide) was observed. Along
with the upward shifting isopycnals, hypoxia ($O_2 < 60$ µmol L$^{-1}$) characterized almost half of the water column at the end of the cruise. Intense hypoxic and even anoxic conditions in the bottom water have been frequently observed at the Boknis Eck Time Series Station (Eckernförde Bay) between late summer and autumn (e.g. Bange et al., 2010; Dale et al., 2011; Lennartz et al., 2014; Ma et al., 2019, 2020).

$CH_4$ concentrations ranged from 4.7 nmol L$^{-1}$ at the surface to 104.0 nmol L$^{-1}$ in the bottom waters (Figure 5B). In contrast to the $O_2$, temperature and salinity profiles, the $CH_4$ distribution in the bottom and intermediate waters showed a larger variability. Strongest accumulation of $CH_4$ was confined to the bottom waters below 20 m. Although the surface samples revealed a stronger oversaturation of $CH_4$ than during the AL510 campaign, the
stratification of the water column seemed to be an effective barrier for the $CH_4$ from the bottom waters reaching the atmosphere.

### 3.3 Seasonal Variability

Between June and September, a strong increase in the salinity, along with changes in the vertical temperature distribution, indicated an exchange of the waters in the western Kiel
Bight over the entire water column (Figure 6). With the change of the water masses, a clear shift in the $CH_4$-$O_2$ relationship from June to September was observed (Figure 7). While only a slight increase in the surface $CH_4$ concentrations was observed between June and September, much higher bottom $CH_4$ concentrations and much lower $O_2$ concentrations were found in September. The co-occurrence of $O_2$ depletion and $CH_4$ enrichment in the bottom
water agrees with the observations from previous studies (e.g. Bange et al., 2010; Steinle et al., 2017).

The high $CH_4$ concentration in the bottom water most likely results from methanogenesis in the anoxic sediments (Bange et al., 2010) producing $CH_4$ that is partly released into the water column (Donis et al., 2017; Reindl & Bolałek, 2014). The summer stratification inhibits the
$CH_4$ from reaching the surface and, thus, $CH_4$ accumulates below the pycnocline. Within the





water column, $CH_4$ is efficiently oxidized and only a small fraction reaches the surface layer (Steinle et al., 2017). The salinity change between June and September also indicates that the high $CH_4$ concentration in the bottom water in September does not result from long-term accumulation of $CH_4$ in the bottom waters. It is rather the result of recent local $CH_4$ release

from the sediments. However, advection of bottom waters that are already enriched in $CH_4$ may additionally contribute to the $CH_4$ distribution.

### 3.4 The 2018 European Heatwave Impact on $CH_4$

Figure 8 shows the anomalies of T and $\Delta CH_4$ in 1 and 25 m water depth from January 2006 to December 2018. A pronounced temperature anomaly is visible in 1 m depth in August

2018 reflecting the heatwave which occurred from mid-July to August 2018 across northwestern Europe (Kueh & Lin, 2020). However, a signal of the 2018 heatwave is not visible in the temperature anomalies in 25 m depth. The maximum anomaly of $\Delta CH_4$ in 1 m water depth is visible in May 2018 and thus not associated with the heatwave signal of the temperature anomaly in 1 m. The maximum temperature anomaly is found in 1 m water

depth for July 2006 and reflects another European heatwave which was experienced by large parts of western and central Europe during July 2006 (Chiriaco et al., 2014). Again, the signal of the 2006 heatwave is not visible in the anomalies of $\Delta CH_4$. Overall, there is no relationship between the water temperature anomalies and $\Delta CH_4$ anomalies in both 1 and 25 m depth.

This finding is in contrast to the results by Borges et al. (2019) who reported significantly enhanced $CH_4$ surface concentrations in coastal waters of the North Sea off Belgium in July 2018. They speculated that the high dissolved $CH_4$ surface concentrations might have been caused by a temperature-driven enhancement of both methanogenesis and sedimentary release of $CH_4$. Humborg et al. (2019) measured dissolved surface $CH_4$ concentrations

during a cruise crossing the northern Baltic Proper between Sweden and Finland after the heatwave in September 2018. They speculated that the heatwave caused higher $CH_4$ emissions to the atmosphere from near shore sites which, in turn, might have been fueled by temperature-driven sedimentary release of $CH_4$. However, our data do not support a heatwave-driven enhancement of $CH_4$ concentrations at Boknis Eck (Eckernförde Bay)

(see Figure 8). Thus, $CH_4$ emissions to the atmosphere at Boknis Eck does not seem to be affected by the heatwaves.

The frequency of higher $\Delta CH_4$ anomalies in 25 m seems to have increased since 2013 (Figure 8). We may, thus, speculate that sedimentary release of $CH_4$ to the overlying water




column may have increased as well which in turn might be caused by the long-term
warming trend observed at Boknis Eck (Lennartz et al., 2014).

### 3.5 CH$_4$ in the Surface Layer

During both cruises, the surface layer was always oversaturated or close to equilibrium with
the atmosphere (Figure 9). $\Delta$CH$_4$ ranged between ~0 and 6 nmol L$^{-1}$ during AL 510 and
between 2 and 8 nmol L$^{-1}$ during AL 516, corresponding to saturations of 103-292 % and 201-
366 %, respectively. The near-surface samples of CH$_4$ from the zodiac and underway
measurements revealed that within the top 1 m of the water column, CH$_4$ concentration
gradients existed (see Figure 10), with larger concentration differences between 0.1 and 1 m
(0.2-2.7 nmol L$^{-1}$) than between 1 and 2 m (0.1 -1.8 nmol L$^{-1}$, mean difference between CH$_4$
gradients: ~1 nmol L$^{-1}$) during the AL 516 cruise. The direction of the gradients was highly
variable, with some stations showing higher CH$_4$ concentrations in the topmost sample, while
others displayed increasing concentrations with depth or intermediate maxima.

Interestingly, the CH$_4$ concentrations measured from the shallowest Niskin bottles (1-2 m)
were generally higher than from the surface samples (average difference between CH$_4$
concentrations from Niskin bottle and from zodiac samples: 1.2 ± 0.4 nmol L$^{-1}$). This could
reflect the different sampling conditions, but it may also be a sign of mixing or carry-over
effects from the CTD profiling. A comparably large variation between the triplicate samples
may result from the challenging sampling conditions on board of the zodiac. However, the
majority of the observed gradients is larger than the cumulative uncertainty of the replicate
measurements.

The sampling depth for surface water is not uniformly defined in oceanic measurements.
While for open ocean CTD sampling depths down to 10 m are recognised as surface
samples, for coastal areas often 1 m is considered as the surface depth (e.g. Ma et al.,
2020). For continuous UW measurements the sampling depth depends on the vessel's hull
and water intake depth which can range between 1 and 10 m depth ( e.g. 2-5 m, Becker,
2016; Karlson et al., 2016; Kitidis et al., 2010; Rhee et al., 2009; Zhang et al., 2014).
Although the near-surface gradients found in our study do not show a clear direction, our
results indicate that at least in coastal areas with elevated CH$_4$ concentrations, a sampling
depth of several meters may not correctly represent the surface CH$_4$ concentration.

## 3. Summary and Conclusions

In order to determine the small-scale variability of dissolved CH$_4$ we set up a purge-and-trap
system with a significantly improved precision. CH$_4$ concentrations were measured with the



purge-and-trap system during two cruises to the western Kiel Bight (incl. Eckernförde Bay) in June and September 2018 as part of the Baltic GasEx study. Special emphasis was put on the sampling of the upper water column and within the mixed layer, with high-resolution
sampling of the top 1 m to determine potential small-scale $CH_4$ concentration gradients within the mixed layer that may not be captured with conventional CTD/rosette-sampling approaches (see e.g. Fischer et al., 2019).

Our measurements revealed higher $CH_4$ concentrations in the bottom waters and comparably lower $CH_4$ concentrations in the surface waters which was caused by a
stratification of the water column. This, in turn, prevented upward mixing of $CH_4$-enriched waters during both campaigns. $CH_4$ concentrations in the bottom waters were significantly higher in September compared to June and were accompanied by a strong decrease in $O_2$ concentrations in the bottom water which led to anoxic conditions favorable for microbial $CH_4$ production in September 2018. The overall setting of the $CH_4$ water column distribution and
the comparably rapid seasonal change in the $CH_4$ concentrations is in line with the time-series measurements of dissolved $CH_4$ concentrations at the Boknis Eck Time Series Station in Eckernförde Bay (Ma et al., 2020; Maltby et al., 2018; Steinle et al., 2017)(Ma et al. 2020; Maltby et al. 2018; Steinle et al., 2017).

In summer 2018, northwestern Europe was experiencing a pronounced heatwave which led
to significantly enhanced water temperatures in the Baltic and the North Seas. This, in turn, might have triggered enhanced $CH_4$ production and consequently might have led to enhanced $CH_4$ concentrations (Borges et al., 2019; Humborg et al., 2019). However, we found no relationship between the anomalies of water temperature and excess $CH_4$ in both the surface and the bottom layer at Boknis Eck (Eckernförde Bay). We conclude that
pronounced European heatwaves which, for example, occurred in 2006 and 2018 did not affect $CH_4$ concentrations in the Eckernförde Bay. Therefore, the 2018 European heatwave most likely had no effect on the observed increase of the $CH_4$ concentrations in the western Kiel Bight from June to September 2018.

$CH_4$ saturation in the surface layer were always >100 % and, thus, the western Kiel Bight
and Eckernförde Bay were sources of $CH_4$ to the atmosphere during both June and September 2018. This agrees with the fact that the Baltic Sea is a source of atmospheric $CH_4$ throughout the year (see e.g. Gülzow et al., 2013; Gutiérrez-Loza et al., 2019; Ma et al., 2020). The high-resolution measurements of the $CH_4$ concentrations in the upper 1 m of the water column were highly variable and showed no uniform decreasing or increasing
gradients with water depth. Surface $CH_4$ concentration measurements used for flux calculations are usually from one depth in the surface layer assuming that there are no concentration gradients and that the $CH_4$ concentration in the surface layer is uniform. Our

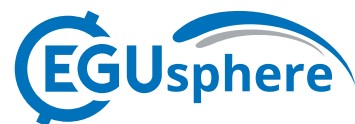

results imply that the assumption of a uniform distribution of $CH_4$ concentrations in the upper
surface layer is not justified. Thus, $CH_4$ flux calculations on the basis of the concentration

difference across the ocean/atmosphere interface is associated with a degree of uncertainty
when ignoring the $CH_4$ variability in the upper surface layer (Calleja et al., 2013; Fischer et
al., 2019). However, since there were no uniform increasing or decreasing $CH_4$ gradients, we
cannot assess whether $CH_4$ flux calculations would have been generally under- or
overestimated.

Overall, our results show that the $CH_4$ distribution in the water column of the western Kiel
Bight and Eckernförde Bay is strongly affected by both large-scale (i.e. seasonal) and small-
scale variabilities. In order to reduce the uncertainties associated with concentration
difference-based $CH_4$ emission estimates, we suggest high-resolution measurements in the
upper surface layer on a regular (at least seasonal) basis.

## Author Contributions


AK, DB and HWB designed the study. SG set up the purge-and-trap system and performed
the measurements. AK, HB and SG and wrote the manuscript. DB carried out the sampling
during both campaigns and contributed to the manuscript.

## Acknowledgements

We thank the captain and crew of R/V Alkor for their support during the Baltic GasEx cruises.
We would like to thank Melf Paulsen, Hanna Campen, and Riel Carlo Ingeniero for their
support with the laboratory equipment as well as Tina Fiedler for the laboratory maintenance
and Xiao Ma for his help with the data acquisition. This work was part of the BONUS
INTEGRAL project which received funding from BONUS (Art 185), funded jointly by the EU,

the German Federal Ministry of Education and Research, the Swedish Research Council
Formas, the Academy of Finland, the Polish National Centre for Research and Development,
and the Estonian Research Council. The Baltic GasEx project was supported by GEOMAR.

## Data Availability

The data is available from the MEMENTO database (https://memento.geomar.de) and from

the PANGEA database under https://doi.org/10.1594/PANGAEA.923992 and
https://doi.org/10.1594/PANGAEA.924372 for AL510 and AL516, respectively. Data from the
Boknis Eck Time-Series Station are available from www.bokniseck.de.



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

## Tables

**Table 1: Boknis Eck April 2020 concentrations and deviations measured with PT and HS.**

| | Purge and Trap | | | Headspace | | |
|---|---|---|---|---|---|---|
| Depth [m] | Mean CH$_4$ [nmol L$^{-1}$] | Std dev [nmol L$^{-1}$] | Std dev [%] | Mean CH$_4$ [nmol L$^{-1}$] | Std dev [nmol L$^{-1}$] | Std dev [%] |
| 1 | 5.89 | 0.12 | 1.99 | 9.26 | 0.84 | 9.06 |
| 5 | 12.97 | 0.25 | 1.89 | 14.99 | 2.16 | 14.42 |
| 10 | 24.61 | 0.56 | 2.27 | 26.55 | 0.63 | 2.36 |
| 15 | 35.24 | 0.09 | 0.25 | 37.46 | 0.30 | 0.80 |
| 20 | 164.17 | 1.08 | 0.66 | 167.57 | 2.34 | 1.11 |
| 25 | 202.50 | 0.37 | 0.18 | 212.93 | 8.91 | 3.80 |



## Figures

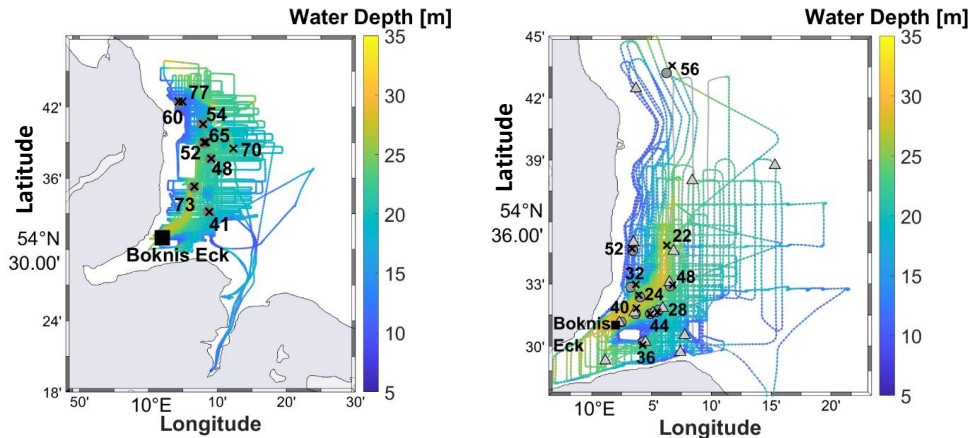

**Figure 1: Cruise tracks of AL510 (left) and AL516 (right). The black crosses mark CTD stations, dark grey dots mark zodiac sampling sites and light grey triangles mark UW sampling sites. All maps that are shown in this work were computed using the m_map toolbox in Matlab (Pawlowicz, 2020).**

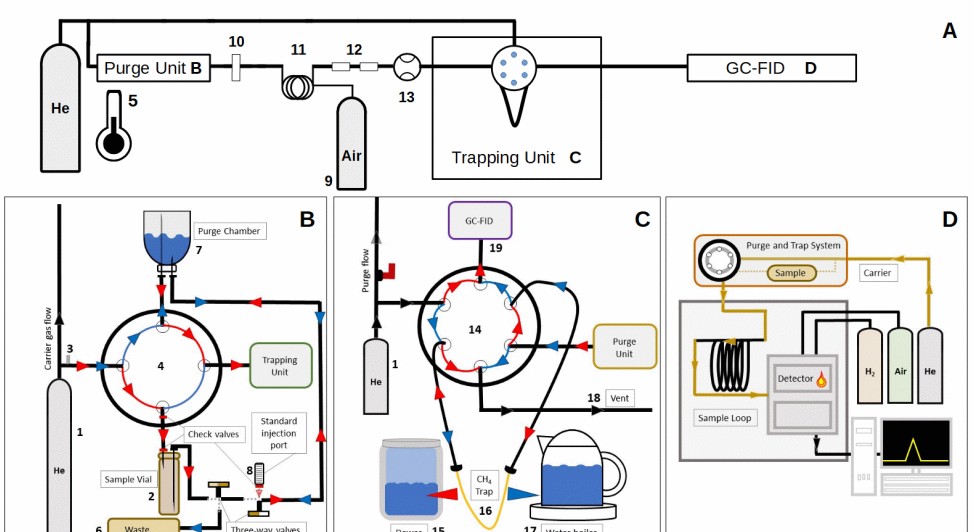

**Figure 2: Schematic illustration of the PT system set-up: A) general set-up of the PT system, the components of the purge unit, the trapping unit and the GC-FID are shown in detail in (A-C); B) set-up of the purge unit showing the direction of the He gas stream when samples are purged (red) and when the purge chamber is emptied (blue); (C) set-up of the trapping unit showing the direction of the He gas stream when the gas is trapped (red) and desorbed (blue); (D) set-up of the GC-FID system. The individual components of the PT system are: 1) He gas cylinder with pressure regulator; 2) sample vial; 3) needle valve; 4) four-port valve; 5) thermometer; 6) double-walled wastewater pipe and wastewater canister; 7) purge chamber; 8) Luer Lock injection port with check valve and Safeflow® infusion valve; 9) compressed air**




cylinder with pressure regulator; 10) filter; 11) Nafion® counterflow drying tube; 12) two glass dry traps filled with $P_2O_5$; 13) flowmeter; 14) VICI Valco® six-port valve; 15) Dewar tank filled with liquid nitrogen; 16) $CH_4$ trap filled with molecular sieve (5Å); 17) water boiler; 18) vent; 19) connection to GC-FID.

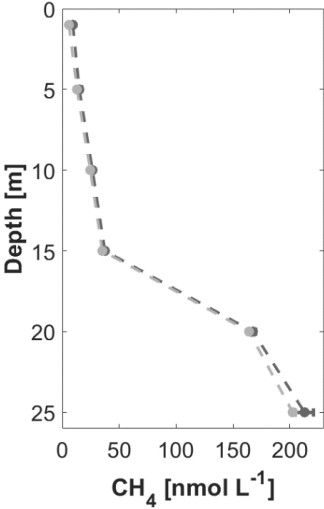

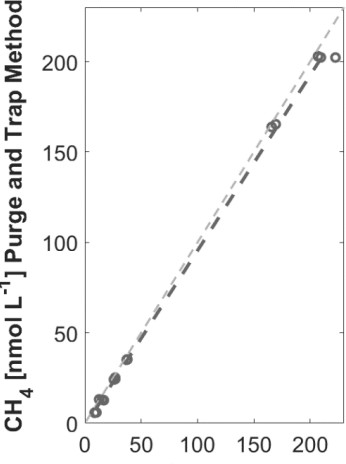

Figure 3a: Boknis Eck depth profile of $CH_4$ measured with PT (light grey) and HS (dark grey) in April 2020. Means are shown as filled dots and dashed line and standard deviation is displayed as error bars in the respective colors.

Figure 3b: Linear regression of $CH_4$ concentrations measured with PT against HS for samples from Boknis Eck in April 2020. The light grey dashed line indicates the 1:1 relation. Y = -1.59 + 0.97 * X. $R^2$ = 0.999. p<0.0001



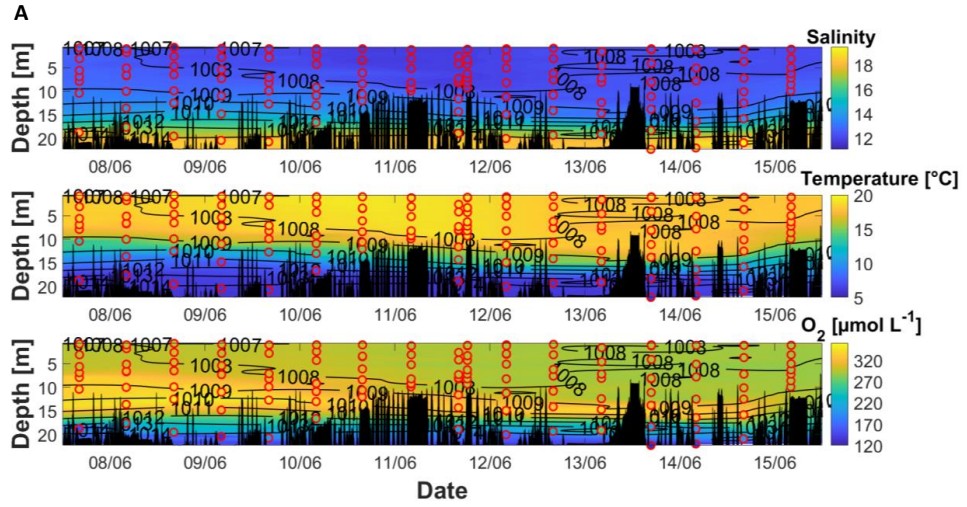

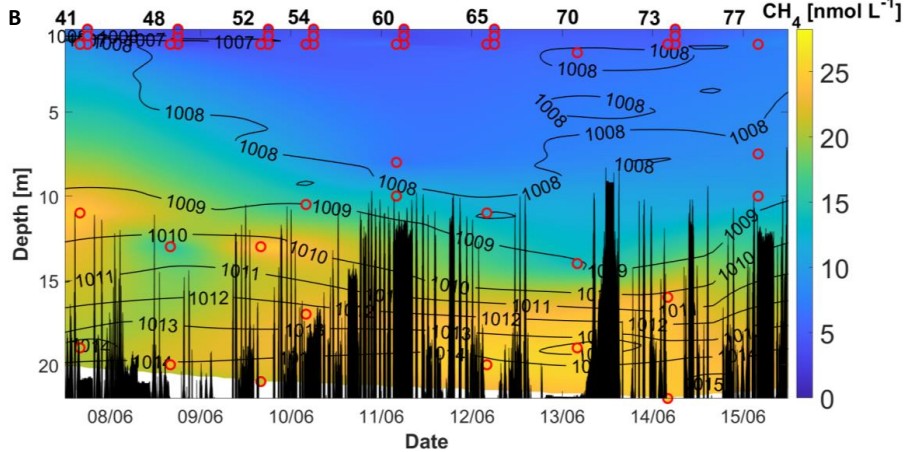

**Figure 4: A) Salinity (upper panel), temperature (middle panel), and O₂ (lower panel) during AL 510 in June 2018. B) CH₄ concentrations during AL 510. Red circles mark the location of the discrete measurements. Black peaks show the topography along the cruise track. Contour lines represent the density. Data for CH₄ was not available for the beginning of the cruise because the measurements started on the 7th June 2018. The station numbers in B refer to those in figure 1.**



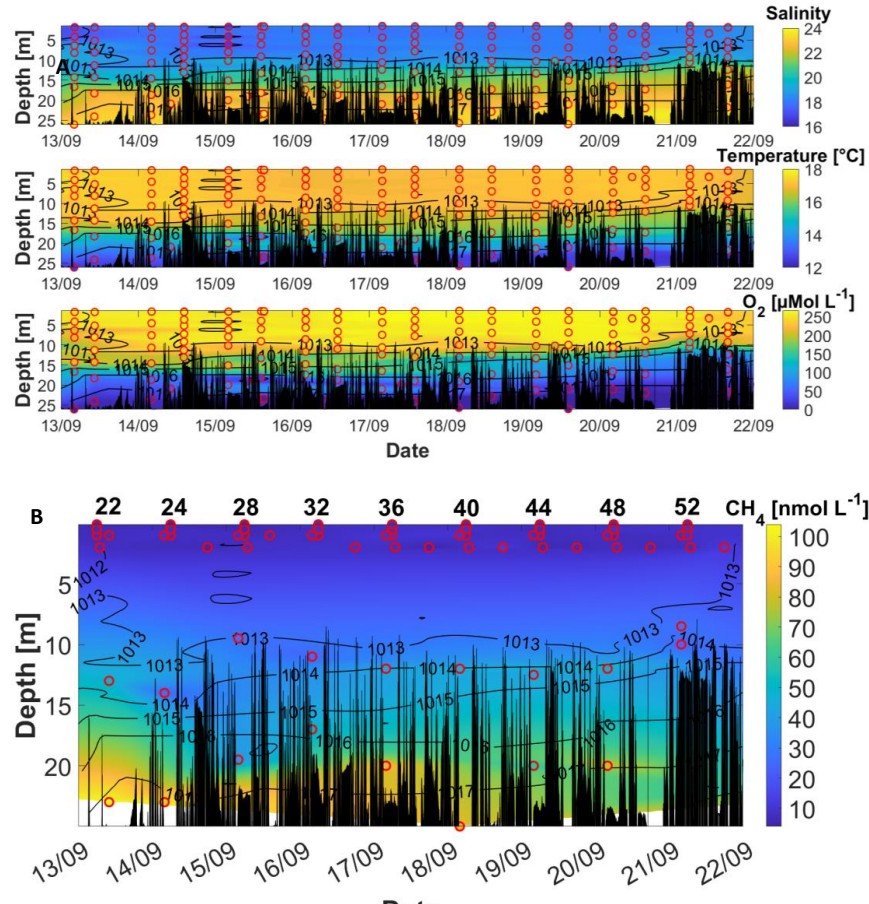

**Figure 5: A) Salinity (upper panel), temperature (middle panel), and O₂ (lower panel) during AL 516 in September 2018. B) CH₄ concentrations during AL 516. Red circles mark the location of the discrete measurements. Black peaks show the topography along the cruise track. Contour lines represent the density. The station numbers in B refer to those in figure 1.**

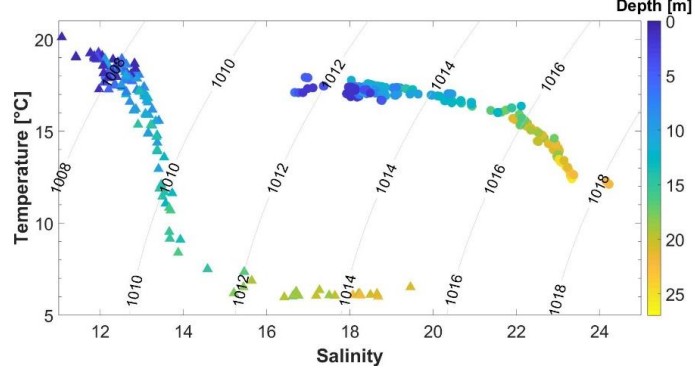





**Figure 6: Temperature/salinity diagram of CTD bottle data from June 2018 (triangles) and September 2018 (circles). Grey lines represent the corresponding isopycnals in kg m$^{-3}$.**

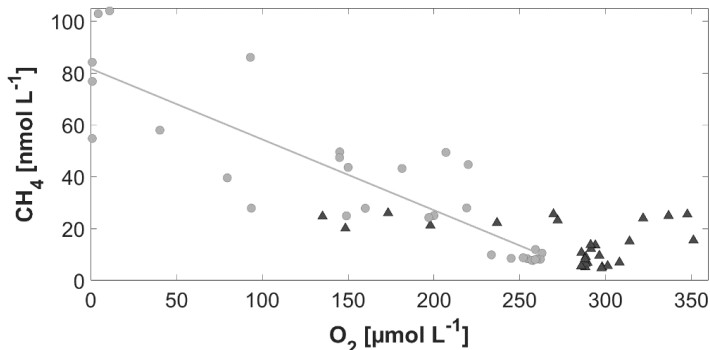

**Figure 7: Comparison of the relationship of CH$_4$ vs. O$_2$ during AL 510 in June (black triangles; p=0.07) and AL 516 in September (filled grey circles, y = -0.2729x + 81.65; R² = 0.764; P=<0.0001).**

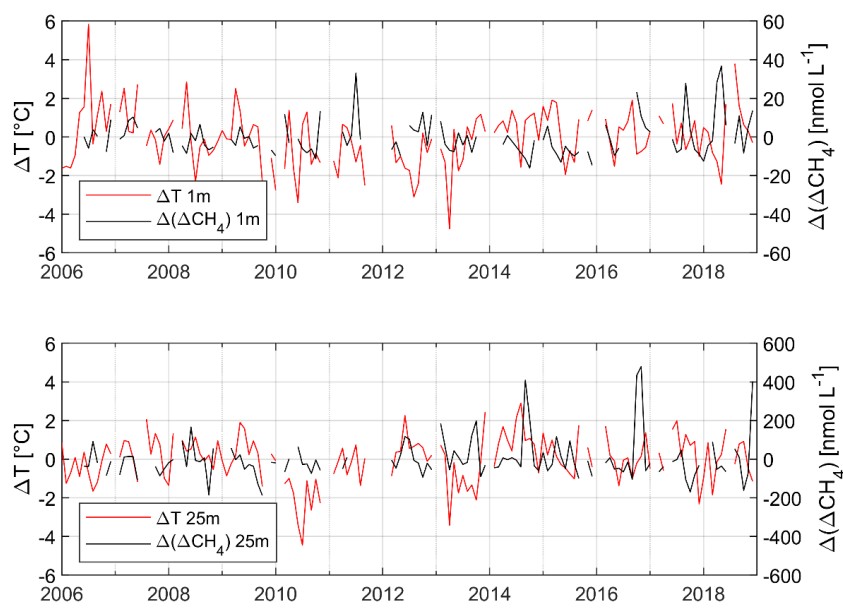

**Figure 8. Monthly anomalies of temperature (□T, red solid line, left y-axis) and □CH$_4$ (□ (□CH$_4$), black solid line, right y-axis) in 1m (upper panel) and 25 m (lower panel) from 2006 to 2019. Please note that gaps in the data sets are caused by missing data.**



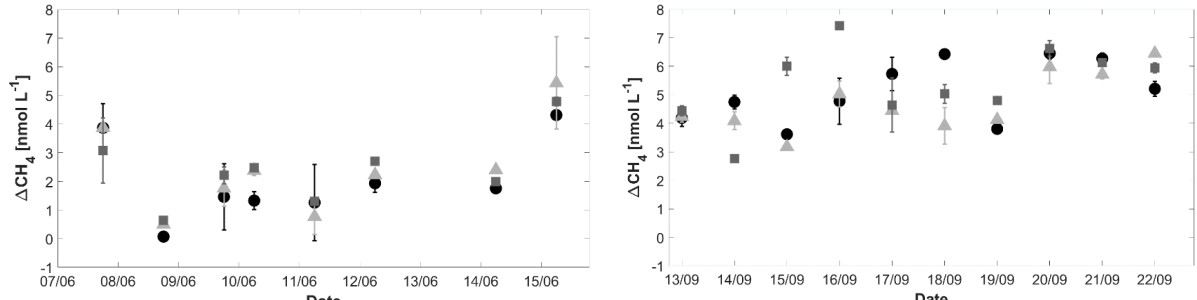

**Figure 9: ΔCH₄ for AL510 (left) and AL516 (right) for surface water at 0.1 m depth (dots), 0.5 m depth (triangles), and 1 m depth (squares). Error bars were calculated as described in the method section.**

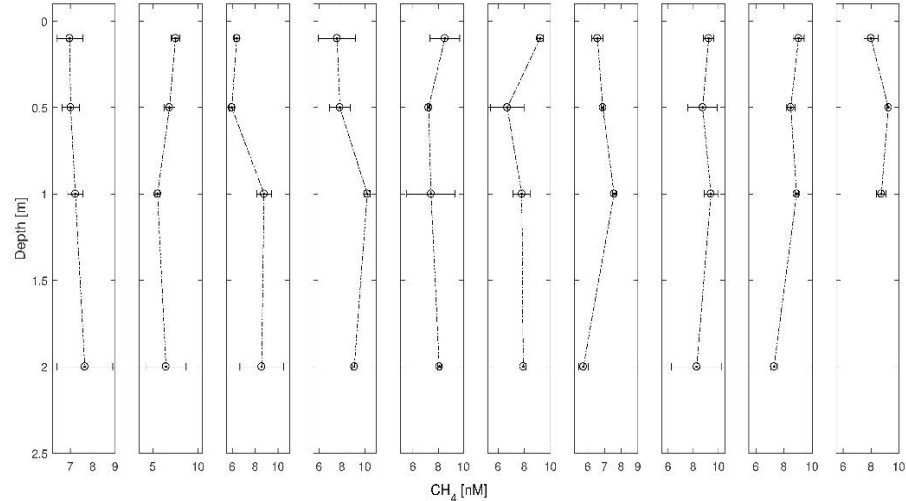

**Figure 10: Near-surface profiles of CH₄ from sampling from the zodiac and UW sampling from the ship (~2 m) during AL 516 in September 2018. The time lag between zodiac sampling and UW sampling was ~1h. All samplings were conducted in the morning (~6:00 UTC). Error bars indicate 95% CI, calculated from triplicate measurements.**