# Peer review of "Seasonal study of the Small-Scale Variability of Dissolved Methane in the western Kiel Bight (Baltic Sea) during the European Heatwave in 2018"

_EGUsphere, 2022_

## Author Comment (AC1)

EGUsphere, referee comment RC1
https://doi.org/10.5194/egusphere-2022-325-RC1, 2022

[Figure]

**Comment on egusphere-2022-325**

Anonymous Referee #1
* * *
Referee comment on "Seasonal study of the Small-Scale Variability of Dissolved Methane in the western Kiel Bight (Baltic Sea) during the European Heat Wave in 2018" by Sonja Gindorf et al., EGUsphere, https://doi.org/10.5194/egusphere-2022-325-RC1, 2022
* * *
**General Comments**

The present paper is dealing with an important scientific topic in marine science: the impact of heatwaves on methane fluxes from coastal waters. It presents a long term temperature and methane dataset combined with a dataset obtained during the summer season 2018 (data includes CH4, T, O2), a year in which northwestern Europe experienced a pronounced heatwave. Studies were conducted in the Kiel Bight and Eckernförde Bay (Balic Sea) and compared with methane datasets from summer 2018 obtained in the North Sea and northern Baltic Proper. In contrast to the last two publications, the present manuscript could not find any significant water column methane enrichments related to the heatwave induced temperature increase (enhanced sedimentary methanogenesis). This makes in my opinion the dataset very valuable, even though the authors do not have any explanation for their differing observation.

The work is complemented by (1) the description of an improved Purge and Trap (P&T) method to measure methane concentrations in high quality and (2) high resolution surface water data to show the inhomogeneity in the distribution within the upper 2 m of the water column. Part (1) is quite interesting, whereas Part (2) is a bit disconnected from the main story (heatwave-CH4), but nevertheless quite interesting. I suggest discussing the high resolution CH4 measurements in context to the methane distribution in the deeper water. Also some parts regarding the surface water sampling in the discussion section could maybe moved to the method section.

Altogether, the work addresses a relevant scientific topic and presents a novel and valuable dataset. The paper is well written, structured and clearly outlined (written in a focused way). The method section (especially the description of the new P&T method) is precisely described and the recent literature cited in the text. The abstract reflects the overall work, the summery seams to my opinion a bit too long and could be shortened (maybe a bit repetitive for such a short paper). The figures are excellent but the authors should take care with the size of the letters, they are sometime difficult to read (e.g. Fig. 10).
Overall, the paper is of good and solid quality and I recommend publication after minor revision.

==AC: Thank you very much.==

**Detailed comments**

**Title**

"Heat Wave" or Heatwave (as written I the following text)?

AC: corrected to 'Heatwave'.

**Introduction**

Line 20
"with a significantly improved precision" Compared to what? Do you mean, compared to other P&T systems described so far in the literature. If so, a direct comparison is missing in the manuscript.

AC: compared to samples measured with static headspace equilibration that has been used in our lab before. A comparison is shown in the method section but we will add the information to this sentence.

Line 44
What about photochemical reactions that leads to the formation of CH4 in surface waters?

AC: Thank you for pointing to the photochemical production. We will add 'A photochemical production of $CH_4$ in oxic surface layers of the coastal and open oceans was suggested only recently as an alternative, non-biological, production pathway (Li et al., 2020)'.
Li, Y., Fichot, C. G., Geng, L., Scarratt, M. G., and Xie, H.: The Contribution of Methane Photoproduction to the Oceanic Methane Paradox, Geophysical Research Letters, 47, e2020GL088362, 2020.

Line 85
Florian Roth et al. (2022, Global Change Biology) showed in a recent paper that a high sampling intensity is required to capture coastal CH4 variability. Do you think it is likely that you missed something with your sampling strategy?

AC: We agree. Indeed, a higher sampling/measurement frequency might have revealed small scale variabilities. Unfortunately, a high-resolution $CH_4$ measurement system (such as laser-based cavity enhanced absorption spectrometer) was not available at the time of our study.

Line 86
I could not find Ho et al. (2019) in the reference list.

AC: Thank you for pointing this out. Ho, D. T., Marandino, C. A., Friedrichs, G., Engel, A, Booge, D., Bange, H. W. , Barthelmess, E.T., Fischer, T., Koffman, T., Lange, F., Quack, B., Paulsen, M., Schlosser, P. and Zhou, L. (2019) Baltic Sea Gas Exchange Experiment (Baltic GasEx). Open Access [Poster] In: SOLAS Open Science Conference 2019. , 21.-25.04.2019, Sapporo, Japan .

Line 94
"higher resolution" What does it mean: mm scale? Not clear at this point.

AC: It is related to sampling the upper 1 m in 3 depths (10, 50 & 100 cm) which is a higher resolution compared to our usual 1 m sample for the surface. In order to avoid any misunderstanding we will delete 'in higher resolution'.

**Methods**

Line 124
Can you be sure that no methane is left in the gas phase of the emptied bottle? Or will the gas phase also be transferred into the purge bottle with the gas stream. Was this tested?

AC: Different tests were performed. 1. We tested calibrations with different purge times between 2.5 and 15 minutes and found the best performance for 4 minutes. 2. We analysed 3 blanks after each standard measurement to make sure blanks were the same as before a standard injection and would not decrease in peak area. 3. On every measurement day we first purged a sample, then ran 3 blanks with that sample remaining in the purge chamber and then injected all our standards for calibration into the filled purge chamber.

Line 135
Why do you need a filter?

AC: Just a first step to stop water droplets from moving into the system

Line 170
You mentioned before that the new P&T is characterized by a "significantly improved precision". Do you mean in comparison to the headspace method? What is the main difference of the new system in comparison to already published systems?

AC: see reply above.

Line 217
„25m water depth" Here and elsewhere (see Fig. 8), please do not forget the space between „25 m".

AC: thank you for pointing this out. This will be corrected in the revised version.

Line 227ff
"ΔCH4 i, avg" is not described in the text.

AC: Thanks for pointing this out. We added the description of "Δ $CH_4$ i, avg " in the text as follows: "Δ (Δ$CH_4$) is calculated similarly to ΔT in the same time period using Δ$CH_4$ which is the monthly excess $CH_4$ (Δ$CH_4$, see above) in 1 (25) m depth. Δ $CH_4$ i, avg is the mean excess $CH_4$ in 1 (25) m depth of the respective month i over this period at Boknis Eck."

Line 284
"oversaturation" is not displayed in Figure 5.

AC: you can see in figure 9 that the surface water Δ$CH_4$ values are positive most of the time which means the surface water was oversaturated with regard to the atmosphere. This would be very hard to read from figure 5 as it only shows concentrations. We will add a reference to figure 9.

**Discussion**

Line 305
"However,…" I do not really understand this sentence. May this needs to be rephrased or your thoughts better explained.

AC: We will delete the sentence ('However, …') and modify the previous sentence: 'It is rather the result of recent local $CH_4$ release either at the BE site itself or advected to the BE site by the bottom waters.

Line 320
Even if it is difficult from your dataset to explain the differences between the two published works and your own observations, it would be interesting to extend the

discussion here: What could be the reason for this discrepancy and the lack of methane enrichment in the Kiel Bight during the heatwave.

AC: We agree and will expand the discussion of the difference between our results and the other studies We will include differences in the stratification regimes (Belgian coastal waters are well mixed throughout the year in contrast to Baltic Sea waters). Moreover, we will include a discussion on how warming will affect both $CH_4$ production and consumption processes which may have led to the low $CH_4$ concentration anomalies at BE.

Line 332ff
Do you mean that the increase in $\Delta CH4$ after 2013 is displayed by the three peaks in Figure 8 in 2014, 2016 and 2018? Or are these peaks related to storm events and the concurrent release of sedimentary methane by methane bubble emissions (Lohrberg et al. 2020, SRep)? Line 336

AC:  Indeed, there are events with very high $CH_4$ concentrations which were attributed to major North Sea water inflow events and sedimentary $CH_4$ release. However, these events occurred in November 2013, March 2014 and December 2014 (see Ma et al., 2020, Lohrberg et al., 2020). The peaks of the CH4 anomaly in the deep waters at BE in 2014, 2016 and 2018 (see Figure 8) do not seem to be related to storm events.

As I mentioned above, this part is interesting but a bit disconnected from the main story. The presented methane concentration data is not discussed in context to the heatwave and the methane data set presented in the deeper water (Figure 4 and 5). Not really sure if this is needed for the manuscript or if this could be better moved to the method section (or supplement).

AC: We agree. We will move Section 3.5 to become the new Section 3.3. With this the connection to the measurements described in Section 3.1 and 3.2 will be highlighted.
Do you have any suggestion how the sampling of surface water should be performed in the future? What kind of process could drive the inhomogeneity of methane concentrations in the uppermost part of the water column (wind, temperature, oxidation,…)?

AC: Future measurements of the surface water should include high-resolution time series measurements of the physical, biological and chemical settings in the upper m of the water column (incl. the so-called surface microlayer). Moored buoys may be a suitable platform to carry senor arrays for time-series measurements. We may speculate that temperature changes can induce small scale stratification events (see Fischer et al., 2019), whereas $CH_4$ production via photochemical production from CDOM may occur in the SML.

Line 346
Are the 2 m samples in Figure 10 taken with a Niskin bottle or UW ship sampling as mentioned in the figure caption?

AC: The 2 m samples were taken from the UW system.

Line 350
Might be good to explain what you mean with "carry over effect".

AC: We closed the bottles during the upcast. That means that while moving up we might bring bottom waters up a bit.

**Summary**
Maybe a bit too long for such a short manuscript. I suggest a shortening of the conclusion.

AC: we will shorten it, see our reply below

Line 365
Is this paragraph really needed in a conclusion?

AC: We will delete the paragraph.

Line 417
„HB" should be „HWB"

AC: Will be corrected.

**Figures**

Figure 1
An overview map of the Baltic Sea is missing.

AC: We will add a map.

Figure 4 and 5
Station numbers are displayed for B but not for A. Why? It might be good to mention that the color scales in Figure 4 and 5 are different.

AC: Station numbers were only needed for methane discussion but will be added to the ancillary parameters as well. We will add one sentence about the different color scales.

Figure 9
Might be better to mention the months instead of the cruise name in the figure caption.

AC: We agree and we will modify the figure captions.

Figure 10
Not sure, if the samples at 2 m water depth were taken with the UW ship sampling and not with the Niskin bottle? The time lag between zodiac and UW sampling is relevant and might have a crucial impact on the methane concentrations. What you see is maybe not indicating a real gradient but rather variability as indicated by your error bars. Size of the letters is too small.

AC:  Samples from 2 m water depth were taken with the UW system. The size of the letters will be increased. The time lag was always within one hour. We will take variability into consideration and rewrite the paragraph.

---

## Author Comment (AC3)

EGUsphere, referee comment RC2
https://doi.org/10.5194/egusphere-2022-325-RC2, 2022

[Figure]

**Comment on egusphere-2022-325**

Anonymous Referee #2
* * *
Referee comment on "Seasonal study of the Small-Scale Variability of Dissolved Methane in the western Kiel Bight (Baltic Sea) during the European Heat Wave in 2018" by Sonja Gindorf et al., EGUsphere, https://doi.org/10.5194/egusphere-2022-325-RC2, 2022
* * *
The authors report a valuable time-series of $CH_4$ in Eckernförde Bay. Such a data-set is precious because long time-series of $CH_4$ are very rare in marine environments. Yet, surprisingly, the data-set of $CH_4$ concentrations shows little inter-annual variations and no clear response to the heat-wave of summer 2018

Is there a variability in salinity during the time-series from 2006 to 2018 (Figure 8) ? Could strong variability of water masses "obscure" signals due to other factors (e.g. heatwave) ?

AC: The $CH_4$ concentration anomaly should be detectable in the surface (5 m water depth) layer. Changes of salinity in the surface layer are usually resulting from mixing events (by storms and/or upwelling). However, mixing of bottom waters brings CH4-enriched waters to the surface and thus results in a pronounced CH4 conc. anomaly in the surface layer (see Ma et al, 2020). However, we could not find a relationship between CH4 surface conc. anomalies and surface temperature anomalies. Therefore, changes in salinities (water masses) are unlikely to obscure the heatwave signals.

While the authors have analyzed inter-annual variations and response to the heatwave of 2018 for $CH_4$ concentration they did not analyze the variability of the fluxes. Would it be possible to compute the fluxes and check if inter-annual changes in wind intensity lead to inter-annual changes of emissions of $CH_4$, even if this is not the case of concentration as suggested by Figure 8 ?

AC: Time series of $CH_4$ flux densities were computed for BE by Ma et al. (2020). Except for a few extremely high flux densities -which resulted from extremely high CH4 surface concentrations in November 2013, February/March 2014 and December 2014- there were no interannual changes (or trends) in the flux densities.

Could it be possible to add information on air temperature close to study site and check if the heatwave of 2018 affected air temperature in the region ? If this is not the case, then it provides an explanation of the absence response of water temperature to the heatwave of 2018. If this is the case, however, it could be useful to try to figure out why the water temperature did not increase in response to a warmer air mass.

AC: There seems to be a misunderstanding. We do not claim to see an 'absence response of water temperature to the heatwave of 2018'. Indeed, we see a strong signal of the heatwaves in the surface water temperatures (e.g. for 2006 and 2018; see Fig 8 and lines 308 – 319).

L110 : what was the delay before analysis ? There could be issues related to long storage of samples (Wilson et al. 2018).

AC: the samples from the Alkor cruises were taken in 2018 and analyzed in 2020. (the samples for the intercomparison of static headspace equilibration and the purge and trap system were analysed within a month after sampling.) The Boknis Eck time series samples were analysed within a few months after sampling. The fact that the $CH_4$ concentrations were supersaturated does imply that there were no problems with contamination with ambient air (via leakage through septa) or preservation (samples have been poisoned and stored in the dark; and thus $CH_4$ production/oxidation were prevented). Please note that the discussion of the storage time in Wilson et al. (2018) is valid for samples with extremely low $CH_4$ concentrations which we did not encounter here.

L320 : Borges et al. also showed that water temperatures were distinctly higher in July 2018 than the previous 14 years. So there was a very distinct increase of water temperature during the 2018 heatwave off the Belgian coast. They also showed a very strong relationship between $CH_4$ and water temperature. I'm unsure the reasoning of Borges et al. can be qualified as "speculation" as stated here. They hypothesized that the response of $CH_4$ was related to higher temperature because it is well established in literature that methanogenesis strongly increases with warming.

AC: In contrast to Borges et al., we could not find a relationship between $CH_4$ concentration anomalies and surface water temperature anomalies at BE. Indeed, any temperature enhancement (i.e. warming) will affect microbial processes. $CH_4$ accumulation is the result of the balance between $CH_4$ production and $CH_4$ consumption processes. Therefore, warming may lead to enhanced $CH_4$ production but also may lead to enhanced $CH_4$ consumption (e.g. via $CH_4$ oxidation). Borges et al. only mentioned $CH_4$ production but did discuss the effect of warming on $CH_4$ consumption. Therefore, their conclusion is speculative because they do not present time series of net $CH_4$ production rate measurements.

However, I suggest to try to discuss the reasons for such a difference. The coastal area studied by Borges et al. is very shallow and does not stratify thermally even in summer (permanently mixed). This might explain the different behavior with Eckernförde Bay where thermal stratification occurs in summer. This leads to a strong physical decoupling between mixed layer and the bottom water and sediments. Such decoupling does not occur in the very shallow area off the Belgian coast, so that warming of surface water

directly impact the bottom sediment, and conversily, enhanced CH4 production in sediments directly propagates to an increase of $CH_4$ concentration in surface waters.

AC: We agree and will expand the discussion of the difference between our results those of Borges at al. We will include differences in the stratification regimes and the warming effect on both $CH_4$ production and consumption (see reply above).

I'm unsure that you can conclude that "Thus, $CH_4$ emissions to the atmosphere at Boknis Eck does not seem to be affected by the heatwaves." The data show that there is no response to the heatwave of 2018. Based on your data you cannot conclude that the site is not affected by heatwaves in general.

AC: We disagree. We do not conclude that BE is not affected by heatwaves. Indeed, there is also a strong temperature anomaly signal for the heatwave in 2006 (see Fig. 8). But this temperature anomaly is (as the signal in 2018) not reflected in the $CH_4$ concentrations (see discussion in lines 314-319). So, our conclusion is that heatwaves are visible in the temp. data set but obviously they do not affect $CH_4$ concentrations and the subsequent $CH_4$ fluxes to the atmosphere during the time period of our study (which spans 13 years).

In summer 2019, there was also a very strong heatwave in Europe that set all-time high temperature records in several EU countries including Germany (Sousa et al. 2020; Vautard et al. 2020). Any chance that CH4 was also measured in 2019 and to add this information to the analysis ?

AC: Indeed, we continue to take monthly samples for $CH_4$ and measurements of the samples from mid 2019 to present are currently ongoing. Please note, however, that the scope of our manuscript is focussed on the results of our study in 2018.

There are several references missing from the reference list.

AC: Thank you for pointing this out. We are sorry for the incomplete reference list. We will correct the reference list.

References

Sousa, P. M., D. Barriopedro, R. García-Herrera, C. Ordóñez, P. M. M. Soares, and R. M. Trigo, 2020: Distinct influences of large-scale circulation and regional feedbacks in two

exceptional 2019 European heatwaves. Commun. Earth Environ., 1, 48,
https://doi.org/10.1038/s43247-020-00048-9.

Vautard, R., et al., 2020: Human contribution to the record-breaking June and July 2019
heatwaves in Western Europe. Environ. Res. Lett., 15, 094077,
https://doi.org/10.1088/1748-9326/aba3d4.

Wilson  et al. (2018). An intercomparison of oceanic methane and nitrous oxide 545
measurements. Biogeosciences, 15(19), 5891–5907. https://doi.org/10.5194/bg-15-
5891-2018

---

## Author Response (AR2)

**Author's response**

One reviewer was wondering in how far your data compare to the 2019 heatwave. I can only speculate that the Bocknis Eco Time series data of that period are used for something else. It would, however, be good to mention in the text that there were other heatwaves in subsequent years and that future studies need to elucidate the effects that these imposed on CH4-dynamics.

REPLY: In the Section 3.5 we discuss that, apart from the 2018 heatwave, heatwaves in previous years did not affect the $CH_4$ anomalies, too. So, our argument is not only based on one single heatwave event. We do not see the point that leaving out the 2019 heatwave is weakening our argumentation and thus the impact of our article.

We added a sentence "Boknis Eck expirienced heatwaves after 2018. However, CH4 concentrations measurements from BE after 2018 were not available at the time of writing of this article." In line 392ff at the end of section 3.5.

The other reviewer mentioned the following points:

Line 292
I think the cruise number is not that important here. Better to mention the respective month "June".

REPLY: we changed all cruise numbers into the respective months.

Line 295ff
Not sure why the cruise numbers are so important here. Better relate the results to the investigated months: June (AL510)- September (AL 516). Please change that (as you did it in 3.4.).

REPLY: we changed all cruise numbers into the respective months.

Line 365
The sentences here "They conclude…" and above (Line 361) "They conclude…" are very similar. I suggest combining the two sentences.

REPLY: These two sentences refer to different studies, therefore, we prefer not to combine them. One starts with they hypothesized and one with they concluded:

"[…] They hypothesized that the high dissolved $CH_4$ surface concentrations might have been caused by a temperature-driven enhancement of both methanogenesis and sedimentary release of $CH_4$. Humborg et al. (2019) measured dissolved $CH_4$ surface concentrations in

the coastal waters of southern Finland after the heatwave in September 2018 (Figure 1). They concluded that the heatwave caused higher $CH_4$ emissions to the atmosphere from near shore sites which, in turn, might have been fueled by temperature-driven sedimentary release of $CH_4$. […]"

Line 386ff
Above you mentioned that the heatwave T-signal was not visible at 25 m water depth. I like the extended discussion here but if the T-signal is not reaching the sediment, it can also not impact the sedimentary microbial methane production.
Are you talking about methane production/consumption in the sediment or water column or both?

REPLY: The significant temperature anomalies associated with heatwaves are indeed not seen in 25 m depth. However, the usual seasonal temperature signal is clearly visible every year in the bottom layer. So, we can expect an impact of temperature on the microbial processes in the sediment.

We refer to both water column and sedimentary processes. We added in Line 380 'in both, the water column and the sediments'.

Line 394
I would delete "gas flares". This is just the acoustic signal. The same in Line 399.

REPLY: We deleted gas flares.

Conclusion
In my opinion still too long. But that might be a matter of taste.

REPLY: We decided leave the Conclusion section as it is.